# Comment on Coleman-DeLuccia Instantons

Tom Banks

Department of Physics and NHETC

Rutgers University, Piscataway, NJ 08854

E-mail: tibanks@ucsc.edu

Bingnan Zhang

Department of Physics and Astronomy,

Rutgers University,Piscataway, NJ 08854

**Abstract**

We complete an old argument that causal diamonds in the crunching region of the Lorentzian continuation of a Coleman-Deluccia instanton for transitions out of de Sitter space have finite area, and provide quantum models consistent with the principle of detailed balance, which can mimic the instanton transition probabilities for the cases where this diamond is larger or smaller than the causal patch of de Sitter space. We review arguments that potentials which do not have a positive energy theorem when the lowest de Sitter minimum is shifted to zero, may not correspond to real models of quantum gravity.

## 1  Introduction

This is a comment on a classic 1980 paper of Coleman and de Luccia[1], whose implications remain widely misunderstood. The aim of the paper was to generalize well understood semiclassical techniques for calculating tunneling probabilities in quantum mechanics and field theory, to an as yet non-existent quantum theory of gravity. This was a daring and fruitful speculation, but the results of the study should be carefully scrutinized to make sure they are consistent with the principles of quantum mechanics.

As an example, the development of the AdS/CFT correspondence has provided us with a convincing collection of models of quantum gravity with AdS boundary conditions. The study of CDL transitions out of an "AdS vacuum"[2][3][4][5][6] has shown that much of the intuition derived from non-gravitational instantons is simply incorrect in this context. We will not review this work, but refer the reader to the excellent references.

This note will instead focus on transitions originating in a "meta-stable" dS space. We will, to a large extent, review older results, and then fill in a few technical details about the

Lorentzian evolution of Big Crunch transitions. Our conclusion will be that there is a large class of Lagrangians, which admit CDL transitions between dS spaces with different radii and dS spaces and a Big Crunch, that are compatible with being a semi-classical description of a quantum system in a finite dimensional Hilbert space, whose maximal entropy is of order the Gibbons-Hawking entropy of the largest dS point. These are models whose potential is *above the Great Divide*: when a constant is added to make the c.c. in the lowest dS minimum vanish, there is no instanton transition because the Minkowski solution obeys the positive energy theorem. We will complete the proof that the maximal causal diamond in the Crunch solution obtained by analytic continuation of the dS instanton always has finite area. When this area is smaller than the area of the dS horizon, then the instanton describes a temporary transition to a low entropy state. The principle of detailed balance then tells us that the reverse transition will occur rapidly in the quantum theory, although there is no semi-classical description of that transition. When the Crunching diamond is larger than the maximal dS diamond, we suggest a quantum model consisting of a large, time dependent, chaotic system, weakly coupled to a smaller system that models a stable dS space. We'll also reiterate the suggestion that Lagrangians below the Great Divide might not be approximations to consistent models of quantum gravity.

Let us begin by reviewing the well understood subject of quantum field theory in a fixed background dS space. This is defined by a Euclidean functional integral on the sphere. Analytically continuing the Schwinger functions on the sphere in one of the Killing coordinates of the sphere, gives us Lorentzian signature Green's functions in a static patch of dS space. These are thermal, with the Gibbons-Hawking temperature. We can then use the dS symmetries to extend the static patch correlation functions to the entire dS manifold. Alternatively, we can analytically continue the polar angle $\theta$ from its hemispherical value $\theta = \pi/2$ and obtain the full dS manifold. By either route, we obtain the Bunch-Davies Greens functions. These have been interpreted[7], following the treatment of Minkowski[8] and AdS[9] black holes, as describing the thermofield double (TFD) state, which purifies the thermal state in the static patch.

It's important to emphasize that the entire region of the Penrose diagram of global dS space1 between any static patch and its TFD image, is causally determined by events in one of the two patches. Despite the fact that global time slices have a spatial volume that increases without bound, the independent canonical degrees of freedom of the model all live in the finite volume spatial patch.

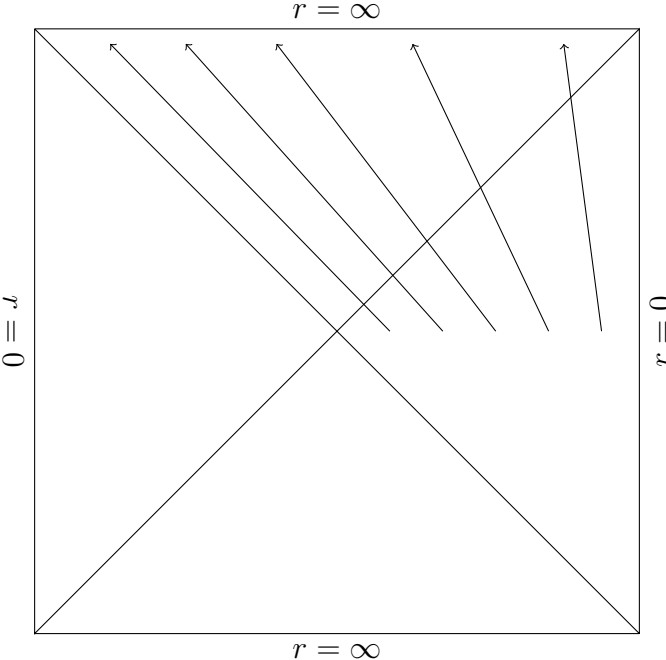

Figure 1: Trajectories of Devices Making Late Time Measurements in QFT in de Sitter Space

This is reflected in the unambiguous predictions of quantum field theory for observations done on this system. Any "measuring device" in quantum field theory is a bound state of quantum field degrees of freedom, which has a large internal Hilbert space and is in a quantum state belonging to a high entropy ensemble of states, each of which has similar values of some set of collective coordinates, including the center of mass of the device. Quantum fluctuations of those coordinates are negligible and coherent quantum superpositions of different histories of collective coordinate expectation values behave like classical probability distributions, with exponential accuracy.

Any such device follows a time-like trajectory and its observations are therefore restricted to a single causal patch. dS QFT predicts a probability distribution for observations made by any such device, which is unambiguous, thermal, and identical for every possible device constructable in dS space. There is nothing that corresponds to the infinities and ambiguities of the theory of eternal inflation. In particular, if the QFT contains scalar fields and a potential with multiple minima, there are unambiguous and identical predictions for the probability that any device will find itself in an environment described by excitations of each of those minima.

In late time global coordinates, we can artificially introduce external sources which probe for the possibility that the field in any small spatial region of a global time slice sits near one of the minima of the potential. The unambiguous QFT predictions for the generating functional in the presence of those sources, is of course dependent on how we choose the time slice, and will contain infinities if we try to make the number of space-like separated sources infinite as we push the slice towards future infinity. These mirror the ambiguities encountered in the theory of Eternal Inflation.

However, if we insist that all of those sources originate as actual physical excitations of quantum fields on the $t = 0$ slice of the standard global coordinates, then the infinities arise only when we take a limit in which the state on that slice is not in the Hilbert space defined

by analytic continuation of the Schwinger functions on the sphere. In addition, the actual disposition of the far future sources is completely determined by the initial conditions, so one cannot do arbitrary things on different time slices in the remote future.

In short, QFT in dS space gives one no grounds for believing in the speculative theory called Eternal Inflation.

## 2  dS to dS transitions

This subject has been thoroughly reviewed before[10][11] so we will be brief. An instanton for a transition out of dS space is a Riemannian manifold with the shape of an ovoid:

$$ds^2 = d\tau^2 + a^2(\tau)d\Omega_3^2. \tag{1}$$

The scalar field satisfies

$$\ddot{\phi} + 3H\dot{\phi} - V'(\phi) = 0. \tag{2}$$

The range of $\tau$ is finite and $a$ vanishes linearly at the ends of the interval. The values of the $\phi$ field at the end of the interval are determined by

$$(\dot{a}_\pm)^2 = 1 - a^2 V \tag{3}$$

Neither of them lies at the minima of $V$, a fact which is attributed to the finite temperature of dS space.

The Euclidean action is negative and one obtains two transition probabilities by subtracting off the negative actions of the two dS points. These probabilities satisfy

$$P_{12} = P_{21}e^{-\Delta S}, \tag{4}$$

where $\Delta S$ is the difference in Gibbons-Hawking entropy of the two dS spaces. This was interpreted in[10] as an expression of the principle of detailed balance. Unitarity implies that every quantum transition is reversible, so that the probabilities of forward and reverse transitions are just the ratios of the number of accessible initial and final states. The fact that the entropy, rather than the free energy appears in the formula for detailed balance, indicates that most of the states have energies below the dS temperature.

The Lorentzian continuation of the instanton can be done from the two points where $a$ and $\dot{\phi}$ vanish, via $a \to ia$ and $\tau = it$. The resulting equations represent open FRW universes, with scalar fields rolling down the potential to their respective dS minima. Brown and Weinberg[11] made the correct interpretation of these solutions as the evolution following the forward and reverse tunneling transitions. They also pointed out that in the limit that the two dS minima were almost degenerate, the scalar field solution approached the instanton for scalar QFT in a fixed dS space. Thus, we should expect that the predictions of quantum gravity approach those of QFT in this limit.

Of course, if one believes in the Covariant Entropy Bound, this cannot be strictly true, because QFT has an infinite number of quantum states in a single causal patch of dS space. Furthermore, an infinite number of those states are localized in the region near the horizon and should contribute a negligible amount to the integral of the stress tensor along any time-like trajectory. The finiteness of the GH entropy suggests that quantum gravity cuts off this infinity

of low energy states. Nonetheless, the qualitative picture of thermal transition probabilities goes over smoothly from the CDL instanton to the QFT limits, when the two minima are close in energy.

Finally, we must mention the Hawking Moss[12] instanton, the dS solution where the scalar field sits at the maximum of $V$. This is the analog of a *sphaleron* solution in flat space QFT, which describes tunneling that is purely thermally activated. The post tunneling Lorentzian evolution is much more complicated. Field theory expanded around a maximum has *many* unstable modes, and localized excitations are more probable than uniform ones. So the prediction of the HM instanton is a complicated probability distribution for different inhomogeneous configurations to roll down the potential. No hair theorems for dS space tell us that the end result of any of these classical solutions looks like empty dS space, but there is no fixed classical picture of the route by which this occurs.

In some cases the HM instanton will have larger transition probabilities than CDL instantons with non-zero $\partial_\tau \phi$, and if the maximum is flat enough, no such CDL instantons exist. However, the action of the HM solution is still negative and the two transition probabilities computed from it still satisfy the law of detailed balance. This is consistent with the unitarity of quantum mechanics. Despite the fact that the post transition history for the HM instanton is predicted only statistically, the no hair theorem assures us that any history will end up in one of the two (or more) dS states. The HM instanton gives us a prediction for the relative probabilities of all of these final states, which is consistent with the general principles of quantum mechanics.

# 3    dS to Crunch Transitions

This section contains the main new result of this paper. For dS to Crunch transitions, the Lorentzian equations are

$$\ddot{\phi} + 3H\dot{\phi} + V'(\phi) = 0. \tag{5}$$

$$H^2 = (\frac{\dot{a}}{a})^2 = \frac{\dot{\phi}^2}{2} + V(\phi) + \frac{1}{a^2}. \tag{6}$$

The system emerges from the tunneling transition at a point where $\dot{\phi} = 0$ and

$$\dot{a}^2 - V(\phi(0))a^2 = 1. \tag{7}$$

This is only possible if $V(\phi_0) < 0$, in which case $a = iV^{-1/2}(\phi(0))\sin(itV^{1/2}(\phi(0))$. The value of $\phi(0)$ is fixed by the instanton solution, of which this is the analytic continuation under $\tau \to it$, $a \to ia$.

We note parenthetically that the case of Hawking-Moss instantons is more subtle and the Lorentzian evolution cannot be obtained by analytic continuation of the instanton. Instead, a typical instability of the configuration at the top of the potential barrier is not homogeneous or isotropic and semi-classical theory predicts at best a probability distribution for different classical motions after tunneling. For fluctuations that return to the dS side of the barrier, the dS no hair theorem assures us that the system will return to its empty dS state. For fluctuations that fall to the negative side of the barrier the situation is much more complicated. We are not aware that it has ever been worked out.

The general picture of post tunneling evolution is clear. The negatively curved FRW universe expands and Hubble friction degrades the kinetic energy of the scalar. At a finite time we reach

a point where $H$ vanishes. However, the derivative of the scalar energy does not vanish at that point, so the system cannot remain there. So generically, $H$ simply changes sign and the universe begins to contract. The scalar energy in a contracting universe increases without bound, and the system does not stay in the basin of attraction of the negative minimum. Eventually, we reach a singularity where $a(t_s) = 0$ and $\dot{\phi}(t_s) = \infty$.

In previous descriptions of these systems, one of the authors (TB) did not consider the possibility that the causal diamond corresponding to the interval $[0, t_s]$ might have infinite area. We now show that this is impossible. A null geodesic in the FRW geometry satisfies

$$\int_0^{r(t)} \frac{dr}{1 - r^2} = \int_0^t \frac{ds}{a(s)}, \tag{8}$$

where the negatively curved spatial metric is

$$ds^2 = \frac{dr^2}{(1 - r^2)^2}. \tag{9}$$

So the spatial coordinate can only go to the boundary if the conformal time goes to infinity. Since we know that coordinate time is in the bounded interval $[0, t_s]$ , this can only happen if $a(s)$ goes to zero. Near $t = 0$, the physical size of causal diamonds goes to zero despite the fact that the past diamond null boundary extends out to spatial infinity. Near the singularity, the scalar kinetic energy dominates the potential, so the equation of state of the scalar is approximately $p = \rho$ and we have $a(t) \sim (t_s - t)^{1/3}$. The coordinate $r(t)$ never gets to the spatial boundary because the conformal time to the singularity is finite.

In previous discussions of this system, TB argued that the rapid variation of $\phi$ would, in the quantum field theory approximation, create a very high entropy state. Eventually the covariant entropy bound would be saturated and one would have to conclude that the maximal entropy was given by the maximal size of the causal diamond. This conclusion is reinforced by our demonstration that the effective equation of state of the Crunch becomes $p = \rho$ near the singularity, since[14] this equation of state saturates the covariant entropy bound.

The conclusion suggested by this analysis is that when the maximal diamond has smaller area than the area of the dS minimum, the CDL transition from dS to a crunch is just a disastrous low entropy transition analogous to all the air in a room collecting in a corner. While the latter situation is fatal for inhabitants of the room, it quickly reverts to the "normal" environmentally friendly equilibrium state.

If we consider potentials "above the Great Divide"[13], where the instanton disappears when we add a negative constant to bring the dS minimum down to zero, then there will be a finite interval in positive vacuum energy for which the dS diamond is larger than the maximal diamond in the crunching region. For these potentials there is a reasonable quantum interpretation of the CDL instanton in terms of a fatal low entropy transition.

What about the opposite situation where the crunching diamond is larger? It's easy to make a quantum model of a finite entropy Big Crunch. Take a Hilbert space whose dimension is equal to the area of the maximal diamond and consider a time dependent Hamiltonian $H(t)$ on the interval $[0, t_s]$. In terms of the generators of the unitary group on the Hilbert space let $H(t) = c_a(t) T_a$ . At $t = 0$ $H$ is some particular unitary operator and can be diagonalized, and if the initial variation of the coefficients $c_a(t)$ is slow compared to typical inverse energy gaps in $H(t)$, we can use the adiabatic approximation to show that there will be only mild changes

in the physics. On the other hand, if after a certain time $t_0$ (corresponding to the point where the classical model flips from expansion to contraction), the time dependence becomes Planck scale, and the algebra generated by the $H(t_i)$ at different Planck times is the whole unitary algebra, then the system will be Planck scale and chaotic. We can now continue to allow time to evolve without ever exiting the Crunch.

Now consider a much smaller finite dimensional system, with a sequence of time dependent operators designed to model a single causal patch in dS space, in diamond universe coordinates, and add a weak coupling to the crunching system, which entangles the two systems on some time scale much longer than the dS Hubble scale. This might be a plausible dual to a dS to Crunch CDL transition when the Crunching diamond is larger than the dS diamond. We can tune the parameters of such a model to reproduce the dS to Crunch transition probability. The system will then obey the law of detailed balance, since it is quantum mechanics. Not much can be said about the detailed time dependence of the coefficients $c_a(t)$ in the Planckian regime. As long as the system is chaotic it will reproduce everything we can guess from the semiclassical CDL equations.

## 3.1   Below the Great Divide

Potentials below the great divide are much more problematic. In ordinary quantum systems, when a meta-stable homogeneous state decays by bubble nucleation, all localized excitations of the meta-stable state simply get swallowed by the expanding bubble of true vacuum. In asymptotically flat space, if an expanding bubble encounters a black hole much larger than itself it clearly gets swallowed by the black hole and the space outside the hole is protected from that particular decay mode. A black hole smaller than the bubble will apparently be swallowed by the bubble, but the area theorem tells us that in classical GR that process must leave over a marginally trapped surface of area no smaller than the black hole horizon. We have seen that the maximal causal diamond inside a crunching bubble has a fixed finite area for a given model, so no matter the apparent external size of the expanding bubble, the process of absorbing a black hole of area comparable to or greater than that maximum, must lead to a major distortion of the bubble itself. We do not have enough intuition about general solutions of GR to know what the answer is. One possibility is a violation of cosmic censorship and the formation of a negative mass naked singularity. A large CDL bubble has large negative energy, much larger in magnitude than the black hole mass. So the object formed when the bubble swallows the black hole definitely has negative energy, as measured at infinity.

One is tempted to say that these classical Lagrangians simply do not correspond to valid quantum models of gravitation. Indeed, there is an enormous amount of evidence from string theory that models of quantum gravity in asymptotically flat space are all exactly supersymmetric and have a positive energy theorem, putting them above the great divide. dS models whose low energy Lagrangian differs from such supersymmetric models by a small positive uplift of the potential will be above the great divide.

On the other hand, any model that modifies the standard model of particle physics by non-gravitational mechanisms such as Casimir energy, producing an apparent instability, is below the great divide because the field potential varies rapidly on the Planck scale[13]. Such Lagrangians may lie in the swampland.

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
