# Peer review of "Comment on Coleman-DeLuccia Instantons"

_SciPost Physics_

## Round 1 · Referee Report · Anonymous (Referee 1) · 2021-8-24

Strengths

This paper is well written, interesting and to the point.

Weaknesses

I think it should be possible to develop the paper's main point somewhat further, cf. comments below

Report

This is a very short paper, a comment really, on the outcome of gravitational tunnelling out of de Sitter in relation to the maximum size of the causal diamond in the crunching region. The paper first reviews old work on this and then adds a note on the above.

The review part is nicely written and from an original angle. The author's stance here is basically that the problems and confusions of eternal inflation are likely to disappear when 1) one considers meaningful observables and 2) one takes in account a model of physical initial conditions. As at least one of the authors knows, this resonates with the quantum cosmology based resolution of the measure problem advocated by Hawking et al. - which does precisely this.

The author's comment tying together the outcome of CdL tunnelling with the max size of the causal diamond in two qualitatively different sets of models, is appreciated. I have one suggestion here, detailed below.

Requested changes

Regards the outcome of the encounter of a CdL bubble with a black hole in models `Below the Great Divide', i.e. the discussion in Sec 3.1. Might a more plausible outcome not be the formation of a black hole with scalar hair? It is precisely in models where there is no positive mass theorem, that the no hair theorem fails. Further, the hair lowers the mass of the hole, as its profile away from the horizon is much like that of a soliton or instanton. Some of this was studied in gr-qc/0608075. I feel this scenario deserves to be added as a possibility and it can perhaps even be studied dynamically, although I don't necessarily suggest the authors do so in this paper.

---

## Round 1 · Referee Report · Anonymous (Referee 2) · 2021-9-22

Report

This paper closes a loose end in a classic problem: the description of CdL bubbles with crunching interior, coming out of the semiclassical decay of a dS spacetime. It is shown that causal diamonds in the bubble's interior remain of finite physical size and support finite entropy. This allows an interpretation of the process as a rare low entropy fluctuation of a putative holographic description of dS as a system with finite entropy.

This is relevant to the general picture developed by one of the authors. Despite its simplicity and its strong reliance on previous papers, I think the question being discussed is of physical importance and I recommend publication.

---

## Editorial Decision

resubmitted